# The Effect of Digestate from Liquid Cow Manure on Spring Wheat Chlorophyll Content, Soil Properties, and Risk of Leaching

Irena Pranckietienė [1], Kęstutis Navickas [2], Kęstutis Venslauskas [2], Darija Jodaugienė [1,*],
Egidijus Buivydas [3], Bronius Žalys [3] and Ilona Vagusevičienė [1]

[1] Faculty of Agronomy, Agriculture Academy, Vytautas Magnus University, K. Donelaičio g. 58, LT-44248 Kaunas, Lithuania

[2] Faculty of Engineering, Agriculture Academy, Vytautas Magnus University, K. Donelaičio g. 58, LT-44248 Kaunas, Lithuania

[3] Lithuanian Energy Institute, Breslaujos g. 3, LT-44403 Kaunas, Lithuania

[*] Correspondence: darija.jodaugiene@vdu.lt; Tel.: +370-37-752-229

**Abstract:** Biogas production from manure and other organic matter, or combinations thereof, is part of the circular economy, and the use of the digestate from biogas production for plant nutrition and thus for soil fertility restoration and organic carbon (C) sequestration completes the circular economy cycle. The use of digestate from biogas production in agriculture is one of the sustainable ways to manage manure or organic waste, an alternative to conventional chemical fertilizers, and a means to achieve the objectives of the European Green Deal. To this end, two two-factor pot experiments were carried out in 2019–2020. Factor A—1. Control (without fertilization), 2. Liquid cow manure 170 kg ha$^{-1}$ N ($N_{170}$), 3. Digested manure 170 kg ha$^{-1}$ N ($N_{170}$), 4. Digested manure 140 kg ha$^{-1}$ N ($N_{140}$), 5. Digested manure 110 kg ha$^{-1}$ N ($N_{110}$); Factor B—1. Plants not cultivated, 2. Plants cultivated. The experiments determined the effect of digestate on the changes in soil nitrate ($NO_3$-N), ammonium ($NH_4$-N) and mineral ($NO_3$-N + $NH_4$-N) nitrogen, and available phosphorus ($P_2O_5$) and potassium ($K_2O$) in the soil without plants as well as with plants cultivated and evaluated the risk of migration of macronutrients into deeper soil layers. The results showed that the application of the highest allowed rates under the Nitrates Directive ($N_{170}$) and $N_{140}$ digestate is accompanied by the addition of high levels of ammonium ($NH_4$-N) nitrogen, which alters the balance of ammonium ($NH_4$-N) and nitrate ($NO_3$-N) nitrogen in the soil, and the extent of their migration to the deeper layers. These results suggest that the application of digestate to agricultural land enhances the migration of nitrogen and, in phosphorus-rich soils, of phosphorus ($PO_4$-P) compounds to deeper layers (>25 cm). In order to achieve environmental objectives, digestate rates should be based on the agrochemical properties of the soil and the needs of the plants and should not exceed 65% of the nitrogen needed by the plants from mineral fertilizers.

**Keywords:** digestate; liquid cow manure; soil properties; nitrogen; chlorophyll content; migration of macronutrients; leaching

## 1. Introduction

Greenhouse gas emissions, nitrate and phosphate enrichment of inland and coastal waters, and deteriorating air quality are major issues throughout the world: increased use of nitrogenous (N) fertilizers in agriculture has significantly altered the global N-cycle [1]; increased use of phosphorus (P) fertilizers—increased eutrophication. Eutrophication has been identified as the primary cause of water quality deterioration in inland waters worldwide, often associated with algal blooms or fish kills [2]. Through better farmland management practices, the emission of nitrogenous gases can be reduced while realizing a clean water environment and climate-smart agriculture [3]. Anaerobic digestion of manures and other organic residues can be used to produce both renewable energy and

organic fertilizer—as biogas and digestate [3]. By employing cost efficiency of energy supply and carbon reduction, the optimization problem addresses the multiple objectives of maximizing energy supply to help improve access to electricity and maximizing carbon reduction to reach the Paris Agreement goals [4]. The use of digestate in plant nutrition is not only very important for maintaining soil fertility, but it is also one of the most sustainable ways to manage manure or organic waste [5]. Anaerobic digestion of animal manure has higher proportions of mineralized plant-available nutrients than untreated manure [6,7]. The concentration of ammonium nitrogen ($NH_4$-N) in digestates provided a good indicator of the nitrogen fertilizer value of the digestates [8]. Higher $NH_4$-N content in digestate is of great importance in fertilizer, as it is immediately available to the plant [9]. The high fertilizing potential of digestate is also associated with a high content of plant-available P and K, and other nutrients. Other problems related to the use of digestates are similar to the use of other types of organic waste-based fertilizers [10–13]: nitrogen mineralization, nitrification, and denitrification have an indirect effect on the environment, as it contributes to the risk of nitrate leaching and losses of $N_2O$ [14,15]: in the high precipitation areas (more than 600 mm average annual precipitation), especially on sandy soils, nitrogen and phosphorus of digestate can be leached into the groundwater and washed into waterways, even after short periods (months and years, rather than decades). Mitigating the vulnerability of agriculture due to capricious weather remains a global challenge [16].

The effect of anaerobic digestates on crop growth after surface application under field conditions is contradictory, since some authors reported higher crop yields compared to undigested slurries [17], whereas others found no effects [18]. Fertilization with biogas digestates positively affects grass yields, but only in single years [6]. Sogn et al. [8] state that digestates often have an invalid ratio between nitrogen, phosphorus, and potassium which is not favorable for plant growth, therefore, the effectiveness of digestate as an organic amendment and fertilizer is still under debate [19].

Managing the risk of nitrate ($NO_3$-N) and phosphorus ($PO_4$-P) leaching is essential for plant fertilization with digestate. Faridullah et al. [20] suggest that reduced leaching of nitrogen and phosphorus from digestate occurs because of the slow release of organic-bound plant nutrients through the mineralization process. However, other researchers point out that digestates supply high amounts of inorganic N that are usually nitrified in a few days [21] and the application of digestate carries a risk of nitrate ions ($NO_3$-N) leaching [22,23].

According to the Nitrates Directive, the amount of nitrogen ($N_{170}$) currently allowed to be incorporated in organic fertilizers must be linked to the mineral nitrogen content of the fertilizer and to the intensity of the transformation of the organic matter during the ammonification and nitrification processes, which determines the amount of nutrients in the soil, the migration of these nutrients and the losses due to leaching.

The aim of the present study is: 1. to investigate the effects of digestate application on the changes of nitrogen (ammonium—$NH_4$-N, nitrate—$NO_3$-N, and mineral ($NO_3$-N + $NH_4$-N)—$N_{min}$ nitrogen), available phosphorus ($P_2O_5$) and potassium ($K_2O$) compounds in the 0–25 cm soil layers with spring wheat (*Triticum aestivum* L.) var. 'Tybalt' $C_2$ and without crops; 2. to evaluate the risk of nutrient (nitrogen, phosphorus and potassium) leaching; 3. to provide knowledge on how to model plant fertilization to optimize plant nitrogen supply through the use of digestate and minimize environmental pollution by nitrogen and phosphorus compounds.

## 2. Materials and Methods

### 2.1. Study Site and Experimental Design

Two-factor pot experiments were set up at Vytautas Magnus University (VMU) Agriculture Academy (Lithuania) in 2019–2020. The soil from the topsoil layer (about 0–30 cm) was used. It was homogenized and moistened to 20% humidity. The soil was characterized by a loam texture (16% of clay, 42.5% of silt, and 41.5% of sand), close to neutral (pH—6.8), low

humus content (1.85%), moderately rich in mineral nitrogen ($N_{min.}$—67.5 kg ha$^{-1}$), high in available phosphorus ($P_2O_5$—227 mg kg$^{-1}$) and available potassium ($K_2O$—281 mg kg$^{-1}$). The experiment consisted of two groups of pots (with spring wheat (*Triticum aestivum* L.) var. 'Tybalt' $C_2$ (1) and without crops (2)).

Two experiments were carried out, which revealed: 1. The effect of digestate on changes in soil nitrate ($NO_3$-N), ammonium ($NH_4$-N), and mineral ($NO_3$-N + $NH_4$-N) ($N_{min.}$) nitrogen and available phosphorus ($P_2O_5$) and available potassium ($K_2O$). 2. The effect of digestate on the leaching of nutrients (nitrogen ($NH_4$-N and $NO_3$-N), phosphorus ($PO_4$-P), and potassium (K)) from the arable land top (0–25 cm) soil layer. In the experiments, liquid cow manure and different rates of digested manure were used according to the nitrogen input per hectare. The experiments in this study included the following treatments: 1. Control (without fertilization), 2. Liquid cow manure 170 kg ha$^{-1}$ N ($N_{170}$), 3. Digested manure 170 kg ha$^{-1}$ N ($N_{170}$), 4. Digested manure 140 kg ha$^{-1}$ N ($N_{140}$), 5. Digested manure 110 kg ha$^{-1}$ N ($N_{110}$). The control rate for liquid cow manure and digestate $N_{170}$ was chosen in accordance with the highest nitrogen limit in the Nitrate Directive. The liquid cow manure and digested manure were incorporated into the soil 7 days before sowing spring wheat in a layer of 0–25 cm (evenly spread over the entire soil volume). On the same day, these fertilizers were also applied to the soil in which no crops were grown. Mineral fertilizers were not used in the experiments. Organic fertilizer has been used once.

The plant growth experiments were performed with spring wheat with 15 plants per pot. Spring wheat was grown in vegetative pots at BBCH 00-25. Treatments were tested in four replicates in a completely randomized block design. Two experiments were conducted and repeated over time in 2019 and 2020.

Pot experiments with spring wheat (*Triticum aestivum* L.) var. 'Tybalt' $C_2$ and without crops were carried out in a greenhouse with regulated interior temperature and humidity. The average air temperature during the experiment was 24 °C during the day and 12–14 °C during the night. The volume of vegetative containers for experiments—1.4 L (plastic pot 8.44 cm in diameter × 25 cm in height). Each pot was filled with 1.6 kg of soil. The density of the formed soil was calculated 1.20 Mg m$^{-3}$. During the experiments, the pots were irrigated regularly to keep the soil water content at about 20%. No water was leached from the pots during the experiment.

After 60 days, the soil was wetted to full soil moisture content to assess the leaching of nitrogen compounds from the soil. The soil was then amended with 200 mL of distilled water at a rate of 50 mL every 4 h over a 12-h period (equivalent to 20 mm of precipitation per day). To prevent soil leaching from the vegetative containers, the bottom of the container was covered with a 1.5 mm diameter plastic net. Leachate was collected using plastic cups.

### 2.2. Preparation of Digestate

The digestate used was obtained from an anaerobic reactor fed with liquid cow manure. The experimental digestates were produced in the Biogas laboratory of Vytautas Magnus University, Lithuania. Anaerobic digestion was performed using a cylindrical continuous type 19 l operating volume biogas digesters, with the procedure described in previous studies [24]. Such digesters are intended for specialized preparation of digestate from several complex feedstocks. The study was performed using an organic load of 3.50–4.50 kg m$^3$ day$^{-1}$ in a mesophilic environment at a temperature of $37 \pm 0.5$ °C.

One week prior to soil application, three digestate and liquid cow manure samples were taken for laboratory analysis. The chemical composition of liquid cow manure and digested manure is given in Table 1. The manure had a dry matter content of less than 8% and is therefore classified as liquid manure. The content of total nitrogen ($N_{sum.}$), available phosphorus ($P_2O_5$) and potassium ($K_2O$) were close to average, while the content of heavy metals did not exceed the content allowed in organic fertilizers [25].

### 2.3. Methods of Analysis

The nutrient content of the soil and the leaching from the soil were assessed 60 days after the application of the digestate, both in the experiment with and without cultivated plants.

Chlorophyll (a, b), carotenoid and total pigments content in leaves were determined at BBCH 25 of spring wheat, using the Holm–Wettstein methodology [26,27].

Liquid cow manure and digestate pH were determined by the potentiometric method in 1N KCl extract; dry matter was determined by drying the sample at 105 °C to constant weight (LST EN 13037:2012); organic carbon was determined by the dry combustion method (LST EN 13037:2012); total nitrogen was determined by the Kjeldahl method (ISO 10694:1995); $N_{min.}$ was determined by the colorimetric method in 1N KCl extract (LST EN 13654-1:2002); available $P_2O_5$ by spectrometric method (LST EN ISO 6878:2004) and $K_2O$ by flame emission spectrometry (LST ISO 9964-3:1998).

Soil texture was determined by sieve-pippete method. Soil texture, was defined by the composition of particle size, namely sand, silt, and clay, according to international soil texture triangle [28]. Soil pH was determined by potentiometric method in 1N KCl extract; organic carbon was determined with Hereaus apparatus; $N_{min.}$ was determined by the colorimetric method in 1N KCl extract (LST EN 13654-1: 2002); $NH_4$-N—with a spectrophotometer (LST EN 13654-1:2002); available $P_2O_5$ and $K_2O$—by Egner-Riem-Doming (A-L) method (LST EN ISO 6878:2004; LST EN ISO 9964:3:1998).

Soil moisture was measured with a Delta-T Devices soil moisture meter *HH2* at a depth of 0–10 cm.

The content of $NO_3$-N in the soil solution was determined calorimetrically; the content of $NH_4$-N was determined with Nesler's reagent; the content of K and $PO_4$-P was determined photometrically.

### 2.4. Statistical Analysis

The statistical analysis of the experimental data was performed by the one-way analysis of variance (ANOVA) using the software *STATISTICA*, version 7. The significances of the differences between the treatments were evaluated using the Fisher least significant difference (LSD) test ($p \le 0.05$). The correlation coefficients and relationships between the $NH_4$-N, $NO_3$-N, and plant photosynthetic parameters (chlorophyll a, b, total pigments, and a/b ratio) ($n_{number\ of\ pairs}$ = 5); between the $NH_4$-N, $NO_3$-N content in the soil and $NH_4$-N, $NO_3$-N content in the leachate ($n_{number\ of\ pairs}$ = 10) tested were determined using the software STATISTICA, version 7 [29,30]. The article presents averages of data from two experiments repeated over time.

### 3. Results and Discussion

### 3.1. Comparison of the Chemical Composition of Liquid Cow Manure and Digestate

Studies show that the mineralization of organic matter during biogas production is more intense than in the field [31,32], resulting in a change in the chemical composition of the final product, the digestate, compared to the original material. The assessment of digestate chemical composition is important for plant nutrition modelling [33], management of leaching of nitrogen and phosphorus compounds [8], and addressing economic [34] and ecological issues. According to the data of this study, the amount of dry matter in the digestate, compared to liquid cow manure, decreased by 28.4%, and organic matter by 32.2 (Table 1). Tambone et al. [35] and Monaco et al. [36] indicate that during anaerobic digestion about 20–95% of the organic matter is decomposed, depending on feedstock composition and anaerobic digestion technology. Fouda et al. [37] and Drosg et al. [38] have reported that the digestate also shows a decrease in total nitrogen ($N_{sum.}$), but Lukehurst et al. [31] argue that the change in total nitrogen can be small.

**Table 1.** Chemical composition of manure and digestate used in the experiment.

| Indicator | Units of Measurement | Liquid Cow Manure | Digestate from Liquid Cow Manure |
|---|---|---|---|
| | | in fresh material | |
| pH | | 8.1 | 8.6 |
| Dry matter | % | 7.42 | 5.31 |
| Organic matter | g kg$^{-1}$ | 58.3 | 39.5 |
| Total nitrogen (N$_{sum}$) | g kg$^{-1}$ | 3.9 | 3.5 |
| Ammonium nitrogen (NH$_4$-N) | g kg$^{-1}$ | 1.5 | 1.9 |
| Nitrate nitrogen (NO$_3$-N) | g kg$^{-1}$ | 0.028 | 0.024 |
| Available phosphorus (P$_2$O$_5$) | g kg$^{-1}$ | 1.7 | 1.8 |
| Available potassium (K$_2$O) | g kg$^{-1}$ | 3.2 | 2.8 |

It is not only N$_{sum}$. nitrogen that is important for plant nutrition, but also NH$_4$-N nitrogen and NO$_3$-N nitrogen, a form of nitrogen that is readily available for uptake by plants. The digestate used in this experiment had 26.7% higher NH$_4$-N content compared to liquid cow manure. Holm-Nielsen and co-authors [39] and Arthurson [40] point out that the increase in NH$_4$-N nitrogen in digestate can be between 10.2 and 70%. This allows plants to use nitrogen more intensively immediately after digestate application or after NH$_4$-N transformation to NO$_3$-N [6,41] compared to manure, but there is a risk of toxicity and loss of NH$_4$-N, especially when the amount of applied NH$_4$-N and NO$_3$-N exceeds the plants' demand, particularly during the early stages of plant development [42].

*3.2. The Effect of Digestate from Liquid Cow Manure on Soil Properties*

In the soil without plants (60 days after incorporation), the NH$_4$-N content did not differ significantly when equal amounts (170 kg ha$^{-1}$) of nitrogen were incorporated with liquid cow manure or digestate or when N$_{140}$ and N$_{110}$ rates of digestate were incorporated (Table 2), but it tended to decrease with decreasing rates of digestate incorporated. The correlation analysis of the data showed that the dependence of NH$_4$-N content on the rate of nitrogen application with digestate was reliable ($R^2$ = 0.881, $p < 0.05$) and very strong. The NH$_4$-N content was 2.24% (liquid cow manure) and 6.77–16.67% (digestate) lower in the spring wheat soil than in the soil without plants, respectively, but not significantly so in all cases. The more intensive N uptake by plants from digestate is also supported by other researchers [31,43], while Siebielec et al. [44] highlight that it also depends on climatic conditions, soil properties, composition of digestate, crop species and time. A significant decrease in NH$_4$-N content in spring wheat was only observed in soil amended with N$_{140}$ and N$_{110}$ digestate rates. This decrease could be due to a more intensive uptake of NH$_4$-N by the plants at lower content of this compound in the soil, as high rates of digestate application may trigger phytotoxic NH$_3$ [45,46]. This confirms the claims of Botheju [47], Van der Eerben [48] and Miller and Cramer [45] that plants are not adequately nourished when high content of NH$_4$-N is present in the soil. The positive effect of lower NH$_4$-N concentrations on plant nutrition is also confirmed by the data from the control variant, which showed a 29.3% reduction in NH$_4$-N in the soil when spring wheat was grown, compared to the amount of NH4-N in the soil without plants.

The highest NO$_3$-N content was found in soil amended with 170 kg ha$^{-1}$ nitrogen with digestate and no crop growth (Table 2). This is significantly higher compared to the soil fertilized with liquid cow manure (N$_{170}$) and the soil fertilized with N$_{140}$ and N$_{110}$ digestate rates. The significantly higher NO$_3$-N content of the soil with the N$_{170}$ digestate application rate could be due to the higher degree of organic matter decomposition and the faster transformation of organic matter into mineral nitrogen compounds compared to manure [49,50]. This data indicate that the application of N$_{170}$ digestate before crop growth and at the right temperature for the transformation of NH$_4$-N nitrogen to NO$_3$-N

(nitrification) nitrogen also increased the risk of nitrogen leaching. Significantly lower $NO_3$-N content was observed in soils amended with $N_{140}$ kg ha$^{-1}$ and $N_{110}$ kg ha$^{-1}$ digestate. These soils had 18.9 and 15.3% lower (significantly) $NO_3$-N content when spring wheat was grown in these soils compared to soils without plants. This leads to the conclusion that both $NH_4$-N and $NO_3$-N nitrogen uptake by the plants was improved by fertilization with digestate at lower than maximum rates.

**Table 2.** The effect of digestate from liquid cow manure on changes in ammonium ($NH_4$-N), nitrate ($NO_3$-N) and mineral ($N_{min.}$) nitrogen in soils.

| Fertilizer | $NH_4$-N mg kg$^{-1}$ | | $NO_3$-N mg kg$^{-1}$ | | $N_{min.}$ mg kg$^{-1}$ | |
|---|---|---|---|---|---|---|
| | Soil without Plants | Soil with Plants | Soil without Plants | Soil with Plants | Soil without Plants | Soil with Plants |
| Control | 2.15c | 1.52a | 15.3b | 13.1a | 17.4b | 14.6a |
| Liquid cow manure $N_{170}$ | 2.23cd | 2.18cd | 16.8c | 15.4bc | 19.0c | 17.5b |
| Digested manure $N_{170}$ | 2.51d | 2.34d | 19.3d | 16.3bc | 21.7d | 18.6bc |
| Digested manure $N_{140}$ | 2.42d | 2.03bc | 16.1bc | 13.0a | 18.5bc | 15.0a |
| Digested manure $N_{110}$ | 2.29cd | 1.91b | 16.2bc | 13.6a | 18.3bc | 15.5a |

Values followed by different letters are significantly different ($p \leq 0.05$) based on Fisher's LSD test.

The highest mineral nitrogen ($N_{min.}$) content 60 days after fertilization was found in the soil receiving the $N_{170}$ kg ha$^{-1}$ digestate application, which was significantly higher than that of the liquid cow manure application or the lower digestate application rates (Table 2). The fact that most of it (88.9%) was $NO_3$-N would predict the highest leaching under excess moisture, but the leaching data show that the amount of $NO_3$-N leached was not significantly different from that from liquid cow manure. Spring wheat growing at BBCH 00-25 on soils fertilized with different rates of digestate showed an average reduction of 15.9% in $N_{min.}$ nitrogen, while a significant reduction in $N_{min.}$ nitrogen was observed in soils amended with $N_{140}$ and $N_{110}$ digestate rates.

The high fertilizing potential of digestate is also associated with a high content of plant-available phosphorus, potassium, and other nutrients [19,51]. However, the nutrient concentrations and the balance between nutrients may vary considerably. Tambone et al. [35] and Bachmann et al. [52] indicate that digestate contains higher amounts of available $P_2O_5$ and $K_2O$ than the parent material. This is confirmed by the data from this study (Table 1).

However, according to Güngör et al. [53], P solubility may decrease after anaerobic digestion due to the formation of struvite and poorly soluble hydroxyapatite compounds. Knowledge of changes in mobile $P_2O_5$ in soil after digestate addition is important because excess amounts lead to eutrophication of freshwater resources [8,54] and it is still unclear how digestate can influence phosphorus availability in soils and consequently $PO_4$-P losses.

In our study, the available $P_2O_5$ content of soils without crops differed insignificantly between liquid cow manure ($N_{170}$) and digestate ($N_{170}$) (Figure 1): 73.9 and 88.9 kg ha$^{-1}$ of available $P_2O_5$ were added in the former and 88.9 kg ha$^{-1}$ in the latter case. This is in line with the findings of Barłóg et al. [55]: "fertilization did not have any significant influence on the content of plant-available phosphorus in soil in any year of the study, regardless of the soil depth, comparing digestate with cattle manure". Similar trends were also found by Vago et al. [56]. In their study, the slight change in available $P_2O_5$ in the soil was probably due to the high pH values of the digestate (Table 1), which led to an increase in the chemical sorption of phosphorus compounds taken up by plants. The lower rates of digestate ($N_{140}$ and $N_{110}$) resulted in a tendentious, but not significantly, lower content of available $P_2O_5$ in the soil compared to the $N_{170}$ digestate rate.

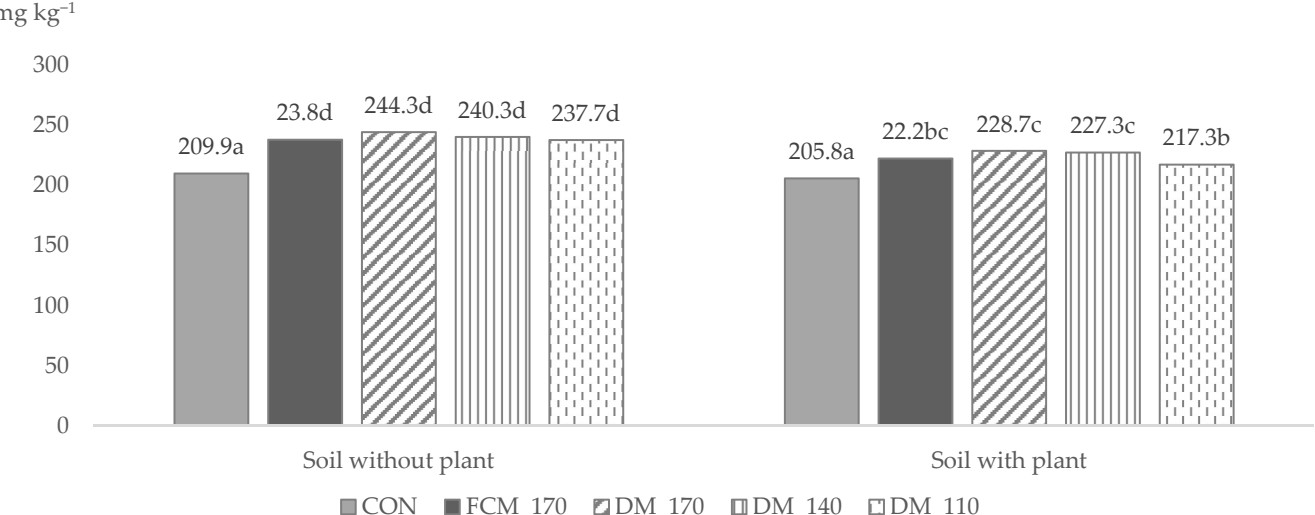

**Figure 1.** The effect of digestate from liquid cow manure on changes in soil content of available phosphorus ($P_2O_5$). CON—control, FCM_170—Liquid cow manure $N_{170}$, DM_170—Digestate $N_{170}$, DM_140—Digestate $N_{140}$, DM_110—Digestate $N_{110}$. Values followed by different letters are significantly different ($p \leq 0.05$) based on Fisher's LSD test.

By assessing the available $P_2O_5$ content of identical soils after plant cultivation, we were able to estimate the uptake of $P_2O_5$ by spring wheat during the first stages of the growing season. It was found that there was no significant difference in the changes of available $P_2O_5$ content in the soil after plant growth at BBCH 00-25, both with liquid cow manure ($N_{170}$) and with digestate at $N_{170}$ and $N_{140}$ (Figure 1). In contrast, the soil amended with the $N_{110}$ digestate rate showed significantly lower content of $N_{110}$ compared to those discussed above: 5.4–8.5% with digestate and 6.6% with liquid cow manure.

Significantly more available $K_2O$ was present in the soil at the highest ($N_{170}$) rate of digestate application (Figure 2). This was due to the higher amount of this element applied to the soil (146.2 kg ha$^{-1}$) with digestate than with liquid cow manure (138.0 kg ha$^{-1}$), although Masse et al. [57] showed that there was no significant difference in the amounts of available K in the parent material (manure) and in the digestate. In contrast, Barłóg et al. [55] state that "in each individual year of the study, fertilization did not exert any significant influence on plant-available K", although the amounts of available K in digestate and cattle manure are different. Lower rates of digestate ($N_{110–140}$) resulted in significantly lower soil available $K_2O$ content compared to $N_{170}$, but no significant differences were found when compared to each other.

### 3.3. The Effect of Digestate from Liquid Cow Manure on Spring Wheat Chlorophyll Content

Plant nutrient supply, especially nitrogen, is characterized by the photosynthetic performance of the plant. Many authors have established that chlorophyll synthesis is dependent upon mineral nutrition because mineral nutrition significantly affects the dynamics of leaf surface formation and the extent of leaf surface, which is reflected in the total sum of leaf surface, the photosynthetic potential, and pure productivity of photosynthesis [58–60].

The analysis of the effect of digestate on chlorophyll changes in the study showed that liquid cow manure and digestate had no significant effect on chlorophyll *b* and carotenoid content, but a significant increase in chlorophyll *a* was observed in the leaves of plants fertilized with $N_{140}$ and $N_{110}$ digestate rates (Table 3). The more appropriate effect of these rates on chlorophyll synthesis in spring wheat was confirmed by the chlorophyll *a/b* ratio. In this respect, the $N_{110}$ rate of digestate had a positive effect.

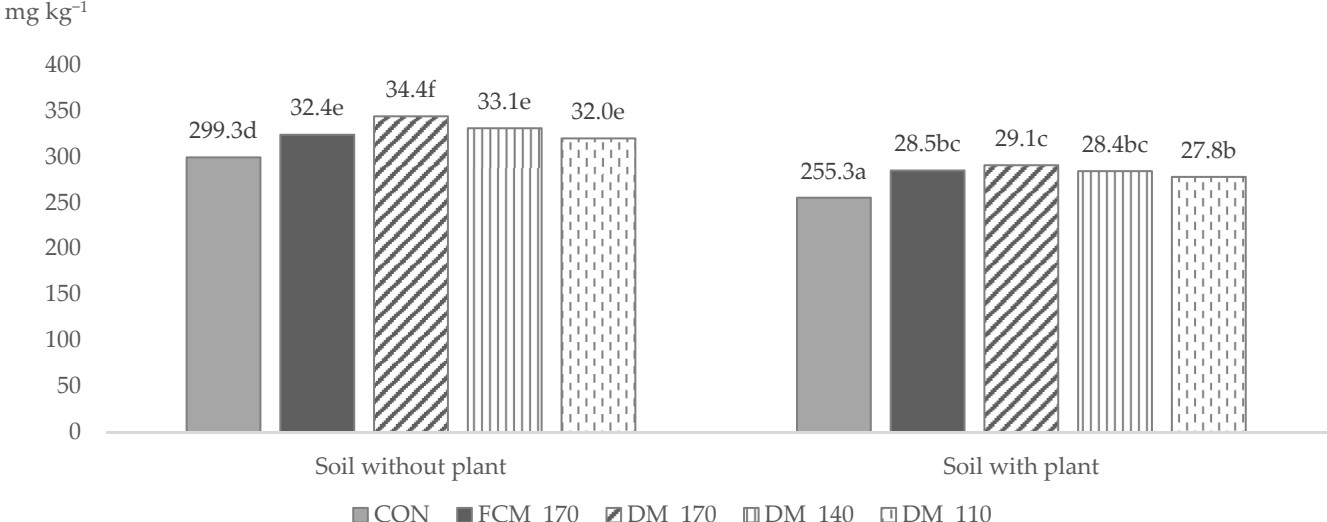

**Figure 2.** The effect of digestate from liquid cow manure on changes in soil content of available potassium (K$_2$O). CON—control, FCM_170—Liquid cow manure N$_{170}$, DM_170—Digestate N$_{170}$, DM_140—Digestate N$_{140}$, DM_110—Digestate N$_{110}$. Values followed by different letters are significantly different ($p \leq 0.05$) based on Fisher's LSD test.

**Table 3.** The effect of digestate from liquid cow manure on photosynthetic parameters of spring wheat.

| Variants | Chlorophyll *a* mg g$^{-1}$ | Chlorophyll *b* mg g$^{-1}$ | Carotenoids mg g$^{-1}$ | Chlorophyll *a/b* ratio |
|---|---|---|---|---|
| Control (without fertilization) | 1.24a | 0.42a | 0.58a | 2.96a |
| Liquid cow manure N$_{170}$ | 1.32a | 0.44a | 0.58a | 3.00a |
| Digestate N$_{170}$ | 1.38a | 0.46b | 0.56a | 2.99a |
| Digestate N$_{140}$ | 1.40b | 0.44a | 0.57a | 3.16b |
| DigestateN$_{110}$ | 1.43b | 0.43a | 0.56a | 3.31c |

Values followed by different letters are significantly different ($p \leq 0.05$) based on Fisher's LSD test.

The correlation analysis showed that the photosynthetic performance of the plants was more influenced by the amount of NH$_4$-N in the soil than by the amount of NO$_3$-N. Positive correlations were found between NH$_4$-N content and chlorophyll *a* ($R^2 = 0.307$, $p < 0.05$), chlorophyll *b* ($R^2 = 0.874$, $p < 0.01$), and total pigment content ($R^2 = 0.467$, $p < 0.05$). NO$_3$-N had a greater effect on the chlorophyll *b* and chlorophyll *a/b* ratio. There was a strong positive correlation of chlorophyll *b* ($R^2 = 0.645$, $p < 0.05$) and a weak negative correlation of chlorophyll *a/b* ($R^2 = 0.195$, $p < 0.05$) with the NO$_3$-N content in the soil.

### 3.4. The Effect of Digestate from Liquid Cow Manure on Leaching of Nutrients

The study shows that increased application of digestate leads not only to positive aspects by improving crop yields and food security, but also to problems of aquatic eutrophication. The dynamics of nutrients after application of digestate to the soil depend on both characteristics [39]. With reference to fertilizer nitrogen use, several processes contribute to the overall efficiency of organic material including digestate applied to the soil, such as ammonia (NH$_3$) volatilization, organic N mineralization, nitrous oxide (N$_2$O) and N$_2$ emissions, and NO$_3$-N leaching. Studies have shown that digestates might lead NH$_4$-N after anaerobic digestion [18]. NH$_4$-N is quickly nitrified to NO$_3$-N, which can be leached [61], especially when large amounts of this compound are added. Many studies have shown that NH$_4$-N nitrogen ions are low in infiltration waters because this form of nitrogen is sorbed by the soil [62]. The results of studies carried out in Lithuania show that the infiltration of NH$_4$-N was only 1.44–1.88 kg ha$^{-1}$ [63]. Our data show a similar trend, with the NH$_4$-N

content of nitrogen ions in the soil leachate from the 0–25 cm soil layer being low, both in and out of crop production, and varying between 0.84 and 1.09 mg $L^{-1}$ (Figure 3). In both cases, the application of liquid cow manure or digestate at a nitrogen rate of 170 kg $ha^{-1}$ did not significantly change the number of $NH_4$-N ions in the soil leachate. This is probably due to the slight difference in the amount of $NH_4$-N applied: 66.9 and 72.5 kg $ha^{-1}$ with liquid cow manure and digestate, respectively. $NH_4$-N ions in the soil leachate tended to decrease with decreasing rates of digestate. The lowest rate of digestate ($N_{110}$) resulted in only a non-significantly higher leaching of $NH_4$-N ions into the soil leachate compared to the control, even though an additional 46.8 kg $ha^{-1}$ of $NH_4$-N was applied with this rate of digestate. The non-significant difference could be due to an increase in the sorption capacity of the soil at the expense of organic matter. This assumption is supported by Ranjbar and Jalali [64] studies: "ammonium adsorption correlated positively with the organic matter content in the soil". Moreover, Al-Saedi et al. [65] reported that "adding organic matter enhanced the adsorption capacity of calcareous soil with an increase in the adsorption of 36%, which indicated that the organic matter is a key limiting parameter in the mechanism of $NH_4$-N adsorption". In most cases, the growth of spring wheat at BBCH 00-25 showed little change in the content of $NH_4$-N ions in the soil leachate, which was directly dependent on the amount of $NH_4$-N applied with fertilizer. There was a positive moderate and reliable relationship between the amount of $NH_4$-N present in the soil and the amount leached into the leachate from the 25 cm soil layer ($R^2 = 0.378$, $p < 0.05$).

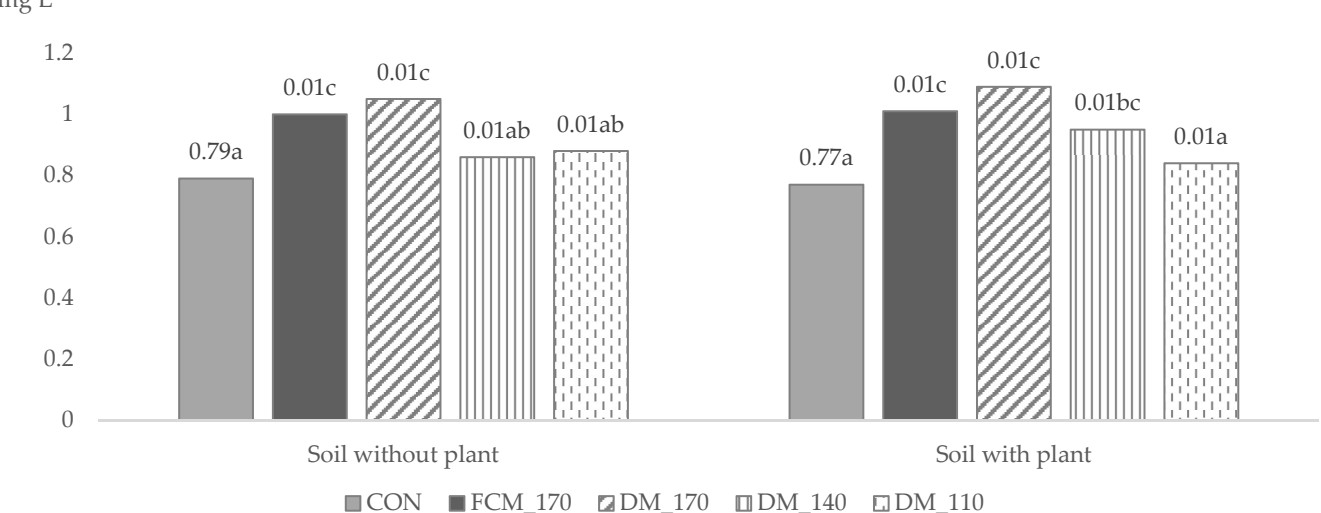

**Figure 3.** The effect of digestate from liquid cow manure on $NH_4$-N in soil leachate. CON—control, FCM_170—Liquid cow manure $N_{170}$, DM_170—Digestate $N_{170}$, DM_140—Digestate $N_{140}$, DM_110—Digestate $N_{110}$. Values followed by different letters are significantly different ($p \leq 0.05$) based on Fisher's LSD test.

Studies show that $NO_3$-N losses depend on year, fertilizer type and location, as well as the amount of nitrogen applied [21,63,66]. According to Svoboda et al. [21], in the short term, digestate has a similar $NO_3$-N leaching potential as liquid cattle manure. The results of this study also confirm that there was no significant difference in the $NO_3$-N ion content of the soil leachate for both liquid cow manure and digestate at the same N application rate (Figure 4). A significant decrease in $NO_3$-N content in soil leachate compared to liquid cow manure and digestate at $N_{170}$ and $N_{140}$ rates was observed when the lowest rate ($N_{110}$) of digestate was applied to the soil at the lowest rate of N, which was equivalent to leaching of the compound from the soil without liquid cow manure or digestate application.

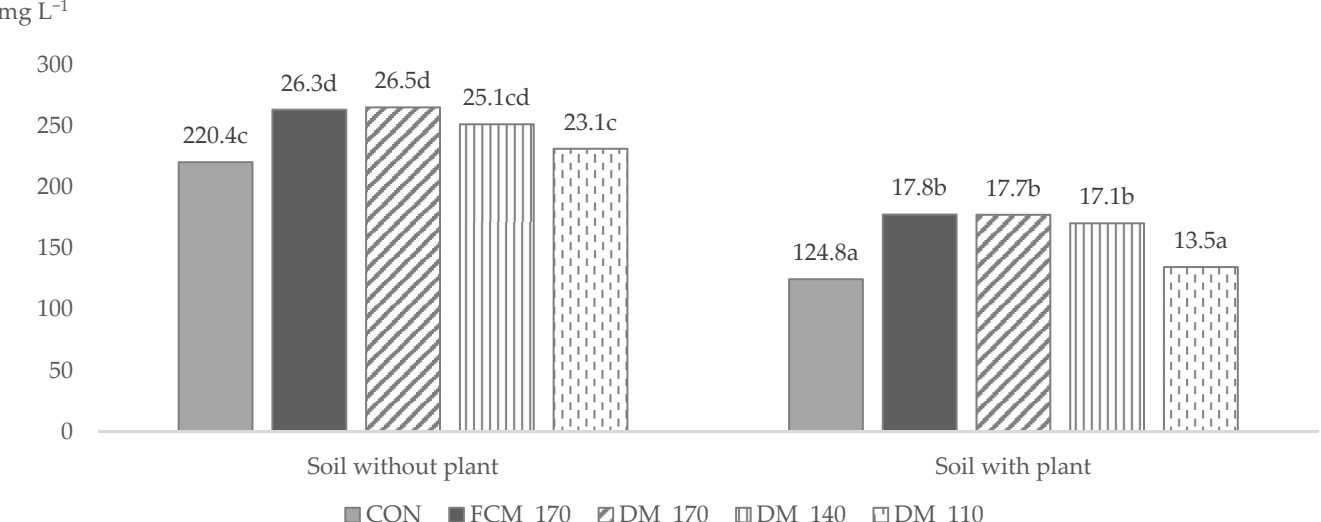

**Figure 4.** The effect of digestate from liquid cow manure on $NO_3$-N in soil leachate. CON—control, FCM_170—Liquid cow manure $N_{170}$, DM_170—Digestate $N_{170}$, DM_140—Digestate $N_{140}$, DM_110—Digestate $N_{110}$. Values followed by different letters are significantly different ($p \leq 0.05$) based on Fisher's LSD test.

Spring wheat grown at BBCH 00-25 reduced on average by a factor of 1.57 the amount of $NO_3$-N nitrogen entering the leachate from the 0–25 cm soil layer. In the soil with plants, the trend of $NO_3$-N nitrogen in the soil leachate followed that of the soil without plants—with decreasing digestate rate their amounts were decreasing. There was a positive and significant correlation between the amount of $NO_3$-N present in the soil and the amount of $NO_3$-N leached into the leachate from the 25 cm soil layer ($R^2 = 0.685$, $p < 0.05$).

In this respect, the rate of digestate application should not exceed the $N_{110}$ limit for nitrogen, even though the nitrate directive allows a limit of $N_{170}$. The advisability of incorporating lower rates of digestate into the soil is also indicated by Tshikalange [67], based on the results of his studies.

In addition to nitrogen (N), potassium (K) and phosphorus (P) are quantitatively the most important plant nutrients, essential to secure proper growth and development. P and K in the feedstock are generally retained during the anaerobic digestion process [68]. Soils overfertilized with phosphorus are a potential risk for the eutrophication of fresh waters through non-point source pollution from agricultural areas [54,69], even though some authors have suggested that phosphorus leaching to groundwater is unimportant because the leaching is negligible [70]. In most cases, phosphorus ($PO_4$-P) concentrations in drainage waters are not high due to slow phosphorus migration, transformation to insoluble compounds, sorption, etc. [71,72]. There are studies showing that the downward movement of $PO_4$-P from organic wastes is potentially significant in areas with coarse soil with low P-absorption capacity, and intensive land use that loads the soil with P either as inorganic fertilizers or organic manures [73–75]. Therefore, it is necessary to study the behavior and interaction of these fertilizers with the phosphorus of soils to evaluate their optimal use and potential environmental problems.

The study showed that 60 days after digestate incorporation into a high phosphorus soil, relatively high amounts of $PO_4$-P were leached from the top (0–25 cm) layer of the soil into the leachate, in the presence of excess soil moisture (Table 3). In the leachate, the soil amended with $N_{170}$, $N_{140}$ and $N_{110}$ digestate rates and no plants was significantly (16.5–27.2%) higher in $PO_4$-P compared to the unfertilized soil, but not significantly different from the liquid cow manure-amended soil. This is supported by the study of Horta and Carneiro [54]. Spring wheat grown at BBCH 00-25 and fertilized with liquid cow manure or digestate at different rates prior to sowing reduced $PO_4$-P content in the soil leachate (20.1

and 11.2–17.6%, respectively) compared to the soil with no plants. However, no significant changes in $PO_4$-P content were found due to different forms and rates of organic fertilizer.

Potassium is another biogenic element that can be leached into deeper layers of the soil or even into drainage or groundwater in the event of excess precipitation, unused by plants and not consumed by the soil [63]. In the present study, the highest leaching of potassium was observed in soils fertilized with liquid cow manure, while substantially lower content was observed in soils fertilized with digestate. When spring wheat was grown, the potassium content of the soil leachate was significantly lower compared to soils without plants (Table 4). Under crop cultivation, potassium content was significantly lower in the leachate of soils fertilized with $N_{140}$ and $N_{110}$ digestate.

**Table 4.** The effect of digestate from liquid cow manure on the content of mobile phosphorus and potassium in soil leachate.

| Fertilizer | $PO_4$-P mg L$^{-1}$ | | K mg L$^{-1}$ | |
|---|---|---|---|---|
| | Soil without Plants | Soil with Plants | Soil without Plants | Soil with Plants |
| Control | 1.21a | 1.26a | 83.9c | 69.9a |
| Liquid cow manure $N_{170}$ | 1.54b | 1.23a | 99.2e | 72.5b |
| Digested manure $N_{170}$ | 1.43b | 1.27a | 94.6d | 73.4b |
| Digested manure $N_{140}$ | 1.41b | 1.24a | 95.8d | 69.7a |
| Digested manure $N_{110}$ | 1.48b | 1.22a | 94.6d | 68.5a |

Values followed by different letters are significantly different ($p \leq 0.05$) based on Fisher's LSD test.

Taken together, the results of the study show that digestate from liquid cow manure is viable as a fertilizer, and that, to achieve environmental objectives, digestate rates should be based on soil agrochemical properties and plant needs but should not exceed 65% of the nitrogen requirement for the target yield. The high $NH_4$-N content in the digestate can result in $NH_4$-N ($NH_3$) toxicity, with negative impacts on growth and biomass production, $NO_3$-N leaching [76]. Good productivity and yield of crops can be achieved if the right amount of N inputs is introduced into the soil system.

## 4. Conclusions

As bio-waste has reached critical levels in many countries, European legislation places the sustainable recycling of bio-waste to address food security, greenhouse gases, and water and soil pollution at the heart of the Global Agriculture towards 2050 policy agenda.

Moreover, agricultural production intensifies, the number of mineral fertilizers used increases, and the percentage of fertilizers not taken up by plants represents a significant economic loss. Under these farming conditions, the possibility of applying energy-saving systems in practice, with the expectation of both economic and environmental benefits, becomes particularly relevant. One solution is the use of digestate from biogas production for plant nutrition, maintaining or restoring soil quality. The demand for digestate in agriculture is linked to the enrichment of the soil with organic matter and nutritional elements, and its proper use in fertilization systems for agricultural crops has great potential.

According to our studies, the application of the maximum rate of digestate allowed under the Nitrates Directive ($N_{170}$) is accompanied by the addition of large amounts of ammonium ($NH_4$-N) nitrogen, which alters the balance of ammonium ($NH_4$-N) and nitrate ($NO_3$-N) nitrogen in the soil and changes the extent to which they migrate to deeper layers.

The results show that intensive fertilization of agricultural land with digestate increases the leaching of nitrogen compounds and, in phosphorus-rich soils, the migration of phosphorus ($PO_4$-P) compounds to deeper layers. To achieve environmental objectives, digestate rates must be based on the agrochemical properties of the soil and the needs of the plants, and therefore the amount of digestate applied should not exceed the $N_{110}$ rate for nitrogen.

Currently, scientific research focuses on the changes in the primary material after biogas extraction, the chemical composition of the digestate obtained, and the effect on plants and soil. As digestate is increasingly used for plant fertilization, it is appropriate to model the composition of the primary material to achieve maximum biogas production and the specific chemical composition of the digestate, without distorting the microbiological processes in the bio-gas reactor and ensuring the stable operation of the power plant. Second, studies on ammonium ($NH_4$-N) nitrogen management during biogas production are also of high relevance.

**Author Contributions:** Conceptualization, I.P. and K.N.; methodology, I.P., K.N., K.V. and D.J.; validation, I.P., K.N. and D.J.; formal analysis, I.P. and K.N.; investigation, I.P., K.N., D.J., B.Ž. and E.B.; resources, I.V.; data curation, D.J. and I.V.; writing—original draft preparation, I.P. and D.J.; writing—review and editing, I.P., D.J. and K.V.; visualization, D.J.; supervision, I.P. and K.V. All authors have read and agreed to the published version of the manuscript.

**Funding:** This research received no external funding.

**Data Availability Statement:** Not applicable.

**Conflicts of Interest:** The authors declare no conflict of interest.

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
