# Peer review of "The Effect of Digestate from Liquid Cow Manure on Spring Wheat Chlorophyll Content, Soil Properties, and Risk of Leaching"

_agronomy, doi:10.3390/agronomy13030626_

Round 1
Reviewer 1 Report
The article entitled “The effect of digestate from liquid cow manure on spring wheat, soil properties, and risk of leaching” is according to the scope of journal.
I am in favor to publish this article if authors consider following changes.
1. The given statement should be upgrade with given studies [1,2] i.e, “Greenhouse gas emissions, nitrate and phosphate enrichment of inland and coastal waters, and deteriorating air quality are major issues throughout the world [1,2]”
[1] Effectiveness of phosphorus control under extreme heatwaves: implications for sediment nutrient releases and greenhouse gas emissions
[2] Estimating smart energy inputs packages using hybrid optimisation technique to mitigate environmental emissions of commercial fish farms
[3] Extreme weather events risk to crop-production and the adaptation of innovative management strategies to mitigate the risk: A retrospective survey of rural Punjab, Pakistan
2. Please write the main research questions at the end of introduction.
3. Please add the theoretical framework of the study before the section 2.
4. In the section of conclusion, policy implications and recommendations for future studies must have to add.
Best of luck with publication!
Author Response
Dear reviewer,
Thank you for your review of our article and for your comments and suggestions.
We provide our responses.
Point 1: The given statement should be upgrade with given studies [1,2] i.e, “Greenhouse gas emissions, nitrate and phosphate enrichment of inland and coastal waters, and deteriorating air quality are major issues throughout the world [1,2]”
[1] Effectiveness of phosphorus control under extreme heatwaves: implications for sediment nutrient releases and greenhouse gas emissions
[2] Estimating smart energy inputs packages using hybrid optimisation technique to mitigate environmental emissions of commercial fish farms
[3] Extreme weather events risk to crop-production and the adaptation of innovative management strategies to mitigate the risk: A retrospective survey of rural Punjab, Pakistan
Response 1: Introduction is corrected and the given statement is upgrade with given studies.
Point 2: Please write the main research questions at the end of introduction
Response 2: The main research questions is corrected and wrote at the end of introduction
Point 3: Please add the theoretical framework of the study before the section 2.
Response 2: Introduction is corrected and the theoretical framework of the study is added.
Point 4: In the section of conclusion, policy implications and recommendations for future studies must have to add
Response 2: The section of conclusion is corrected.

Reviewer 2 Report
1. Abstract should be improved and add year of experiment conducted.
2. Chemical composition of manure and digestate is on dry weight basis or fresh, please clarify??
3. Author conducted study in pot experiment for one year only, and it is not sufficient for any conclusion or recommendation.
4. Maintain uniformity in writing the references.
Overall, efforts of authors are good but paper may accept in ‘Short Communication’ only after revision of the paper as per comments.

Author Response
Dear reviewer,
Thank you for your review of our article and for your comments and suggestions.
We provide our responses.
Point 1: Abstract should be improved and add year of experiment conducted
Response 1: Abstract is improved and added year of experiment conducted.
Point 2: Chemical composition of manure and digestate is on dry weight basis or fresh, please clarify?
Response 2: The chemical composition of manure and digestate was provided in Table 2. in natural material. Now is corrected “in fresh material”
Point 3: Author conducted study in pot experiment for one year only, and it is not sufficient for any conclusion or recommendation
Response 2: The study in pot experiment was conducted for two year (2019 and 2020). In the section "Statistical analysis" was wrote "The article presents averages of data from two experiments repeated over time."
Point 4: Maintain uniformity in writing the references.
Response 2: Reference list we made with program RefWorks and hadn't noticed that it is not uniform. Now we corrected. Thank you!

Reviewer 3 Report
The work presents the important issue of proper nutrient management from various organic fertilizers, especially in terms of the risk of nutrient leaching under conditions of soil water leachate. The presented topic of the manuscript refers to the presented content of the work. The following comments should be included in the manuscript:
In the materials and methods chapter lacks information on how many replicates each fertilizer treatment was carried out
The methodology states that the content of phosphorus potassium in fertilizers was determined, but no information is given on what forms of both elements were determined, whether total or available forms
There are errors in the manuscript of the given No. of methods of determining nutrients
In the manuscript there is an inaccuracy in the use of names, in the abstract "fresh manure'' is used and in the description in Table 2 "liquid manure'' is used, in the literature the word "slurry'' is used more often
In the introduction it is stated that the purpose of the study is to determine the effect of different organic fertilizers on changes in the content of both mineral forms in the soil, in Table 2 there is no data of analysis of the content of nitrate nitrogen, N-NO3, in fertilizers
In Table 2 there is no information whether the content of components refers to the content of dry matter or fresh matter, the content of components in fertilizers is given in %, nowadays in the literature units of g∙kg-1 are used
The manuscript does not maintain the same style of naming, in line 190 or 200 "manure" is used instead of “liquid manure” or "slurry", manure and slurry are different fertilizers with different properties
In lines 199-201 “the NH4-N content was 2.24% (manure) and 6.77–16.67% (digestate) lower in the spring wheat soil than in the no-till soil,…” the use of no-till soil sounds very strange and there is no justification to describe a control soil without plants in a pot experiment, no-till is one of the soil cultivation systems
the statement made in lines 216-219 “The significantly higher NO3-N content of the soil with the N170 digestate application rate could be due to the higher degree of organic matter decomposition and the faster transformation of organic matter into mineral nitrogen compounds compared to manure…” has no justification, due to the lack of information on the nitrate nitrogen content of the analyzed fertilizers
Author Response
Dear reviewer,
Thank you for your very careful review of our article and for your comments and suggestions.
Point 1: In the materials and methods chapter lacks information on how many replicates each fertilizer treatment was carried out.
Response 1: In the section of materials and methods we have provide additional information about replication of each fertilizers treatment.“ Organic fertilizer has been used once. Mineral fertilizers were not used in the experiments.”
Point 2: The methodology states that the content of phosphorus potassium in fertilizers was determined, but no information is given on what forms of both elements were determined, whether total or available forms
Response 2: The chemical composition of manure and digestate was provided in Table 1. There was provided content of “mobile phosphorus and mobile potasium”. Now is corrected “available phosphorus and available potasium” in the table and in all manuscript.
Point 3: There are errors in the manuscript of the given No. of methods of determining nutrients
Response 3: It is now corrected.
Point 4: In the manuscript there is an inaccuracy in the use of names, in the abstract "fresh manure'' is used and in the description in Table 2 "liquid manure'' is used, in the literature the word "slurry'' is used more often
Response 4: It is now corrected and presented uniformly. Liquid cow manure or digestate from liquid cow manure was used in our research. Other expressions can be used when comparing with the data of other authors, for exaple "cattle manure" and so on.
Point 5: In the introduction it is stated that the purpose of the study is to determine the effect of different organic fertilizers on changes in the content of both mineral forms in the soil, in Table 2 there is no data of analysis of the content of nitrate nitrogen, N-NO3, in fertilizers
Response 5: It might be you had in mind Table 1? It is now corrected and provided the data of the content of nitrate nitrogen in the Table 1.
Point 6: In Table 2 there is no information whether the content of components refers to the content of dry matter or fresh matter, the content of components in fertilizers is given in %, nowadays in the literature units of g∙kg-1 are used
Response 6: It is now corrected.
Point 7: The manuscript does not maintain the same style of naming, in line 190 or 200 "manure" is used instead of “liquid manure” or "slurry", manure and slurry are different fertilizers with different properties
Response 7: It is now corrected and presented uniformly.
Point 8: In lines 199-201 “the NH4-N content was 2.24% (manure) and 6.77–16.67% (digestate) lower in the spring wheat soil than in the no-till soil,…” the use of no-till soil sounds very strange and there is no justification to describe a control soil without plants in a pot experiment, no-till is one of the soil cultivation systems
Response 8: Sorry, this was a translation error. It is now corrected and presented “The NH4-N content was 2.24% (liquid cow manure) and 6.77–16.67% (digestate) lower in the spring wheat soil than in the soil without plants, respectively, but not significantly so in all cases.”
Point 9: The statement made in lines 216-219 “The significantly higher NO3-N content of the soil with the N170 digestate application rate could be due to the higher degree of organic matter decomposition and the faster transformation of organic matter into mineral nitrogen compounds compared to manure…” has no justification, due to the lack of information on the nitrate nitrogen content of the analyzed fertilizer
Response 9: It is now corrected and provided the data of the content of nitrate nitrogen in the Table 1.
Thank you !

Reviewer 4 Report
I am writing to you regard the manuscript entitled: The effect of digestate from liquid cow manure on spring wheat, soil properties, and risk of leaching
I think it is lies within the scope of the Journal. However. I have some suggestions to improve it more.
The abstract should state briefly the purpose of the research, applied methods, the principal results, and major conclusions. There is no explanation about the materials and methods, the results of wheat plant cultivation, and the analyzed parameters including total pigments, carotenoids, and chlorophyll I and II. The abstract should be thoroughly revised and improved. In my opinion similar researches have been previously done by others. It is essential to mention the novelty and importance of the study in introduction section.
Page 1: More suitable title should be selected for the article, the authors should specify in the title which feature of the wheat plant has been studied upon the incorporation of digestate
Page 1, line 14: What does the letter (C) stand for? If it stands for (carbon) and is mentioned for the first time, it should be written as carbon (C).
Page 1, lines 18 and 19: Abbreviations ( N140, N110, N170) should be explained. Please clarify and explain in more detail either in the abstract or in the materials and methods section.
Page 1, line 20, 21: (NO3-N), (NH4-N) are ion forms of nitrogen and should be changed to NH4+-N and NO3--N throughout the article.
Page 1, line 21: What is the exact concept of mineral nitrogen in the abstract? Please clarify.
Page 1 and 2, lines 34-77: The literature review is not enough. It is critical to update the MS by adding some recent publications and making a stronger argument about the novelty of the paper. Furthermore, in the introduction section, the words phosphorus and nitrogen should be written in full for the first time and then in the form of abbreviations (N and P). Please correct this issue throughout the article
Page 2, line 71: Which nutrients does the term ((unwanted nutrient leaching)) include?
Page 2, lines 80-87: The results of soil properties should be reported in the results and discussion section. In addition, the authors should clarify the fraction of measured phosphorus and potassium in soil (Total P or available or …)?
Page 2, line 81: Was the studied soil sampled from arable land?
Page 3, line 108: What is the concept of (1.41)? The units of 1.41 must be referred to in the text.
Page 3, line 110; the authors should present the methods used for soil density measurement in section 2.3.
Page 3, line 135: Table 1 should be placed in the results and discussion section not in materials and methods. Please correct.
Page 4, line 170: This section is very long and difficult to read because of the lack of sub-headings. For example, from lines 171 to 193, the paragraph talks about the comparison of cow liquid manure and digestate, and it has no relevance to the title. This paragraph can independently have separate sub-headings.
Page 4. Line 147: Based on this sentence, the authors have determined inorganic nitrogen but reported only ammonium content in table 1. Please correct and clarify this contradiction.
Page 4. Line 148: The authors should briefly mention the method used for phosphorus and potassium measurement.
Page 4, Line 149: Based on this section, the authors should add the applied method for soil texture measurement earlier in the material and methods section. In addition, the content of soil organic matter should be reported.
Line 14, 317, and 165: The authors have reported the content of total pigments and carotenoids in the aforementioned lines, but no explanation has been provided about the applied methods. The relevant method should be presented in the materials and methods section.
Line 168: There is a contradiction between line 168 and line 104, Based on line 104, the test was performed in 4 replicates, and authors should correct and clarify this section.
Page 4, Line 171-184: This section should be revised and be better organized. The authors should first report the experimental results of this study and improve this section with more involved mechanisms. Then, compare the results of the present research with some similar studies, which is done before.
Page 5, Line 219: Provide a reference to this sentence, please.
Figures 1, 2, 3, and 4: The determinations of these figures have to include the error bars.
Table 4: (( PO4)) needs to be corrected in table 4
Author Response
Dear reviewer,
Thank you for your very careful review of our article and for your comments, corrections, and suggestions.
We provide our responses:
Point 1: The abstract should state briefly the purpose of the research, applied methods, the principal results, and major conclusions. There is no explanation about the materials and methods, the results of wheat plant cultivation, and the analyzed parameters including total pigments, carotenoids, and chlorophyll I and II. The abstract should be thoroughly revised and improved. In my opinion similar researches have been previously done by others. It is essential to mention the novelty and importance of the study in introduction section
Response 1: The abstract has been slightly corrected and supplemented. However, we cannot expand much, because according to the requirements, it should not exceed more than 200 words.
Point 2: Page 1: More suitable title should be selected for the article, the authors should specify in the title which feature of the wheat plant has been studied upon the incorporation of digestate.
Response 2: Thank you for the your suggestion, we have been slightly corrected
Point 3: Page 1, line 14: What does the letter (C) stand for? If it stands for (carbon) and is mentioned for the first time, it should be written as carbon (C)
Response 3: It is corrected.
Point 4: Page 1, lines 18 and 19: Abbreviations (N140, N110, N170) should be explained. Please clarify and explain in more detail either in the abstract or in the materials and methods section.
Response 4: It is corrected. N170 changed to 170 kg ha-1 N; N140 changed to 140 kg ha-1 N; N110 changed to 110 kg ha-1 N;
Point 5: Page 1, line 20, 21: (NO3-N), (NH4-N) are ion forms of nitrogen and should be changed to NH4+-N and NO3--N throughout the article.
Response 5: We noticed that in different sources write both, for example NO3-N or NO3--N. We unified whole text and left it NO3-N and so on.
Point 6: Page 1, line 21: What is the exact concept of mineral nitrogen in the abstract? Please clarify.
Response 6: It is corrected. “… mineral (Nmin.) nitrogen” changed to „mineral (NH4-N + NO3-N) nitrogen”.
Point 7: Page 1 and 2, lines 34-77: The literature review is not enough. It is critical to update the MS by adding some recent publications and making a stronger argument about the novelty of the paper. Furthermore, in the introduction section, the words phosphorus and nitrogen should be written in full for the first time and then in the form of abbreviations (N and P). Please correct this issue throughout the article.
Response 7: 1. Introduction is corrected and supplemented.
Point 8: Page 2, line 71: Which nutrients does the term ((unwanted nutrient leaching)) include?
Response 8: “... without crops and to evaluate the risk of unwanted nutrient leaching” changed to „without crops and to evaluate the risk of nutrient (nitrogen, phosphorus and potasium) leaching”.
Point 9: Page 2, lines 80-87: The results of soil properties should be reported in the results and discussion section. In addition, the authors should clarify the fraction of measured phosphorus and potassium in soil (Total P or available or …)?
Response 9: We have provided P2O5 or K2O, it means available. However, we have corrected and added additional information for clarity.
Point 10: Page 2, line 81: Was the studied soil sampled from arable land?
Response 10: Yes, the studied soil sampled from arable land. In the article we provided additional information.
Point 11: Page 3, line 108: What is the concept of (1.41)? The units of 1.41 must be referred to in the text.
Response 11: It was misunderstanding. It was indicated the units, but the letter "l" is similar to “1” (one). We changed and wrote "L", which means liter.
Point 12: Page 3, line 110; the authors should present the methods used for soil density measurement in section 2.3.
Response 12: Soil density is calculated. To form the density of the soil, you need to multiply the volume of the pot by the desired density. In this way, the required amount of substrate is known. In the article is corrected “The density of the formed soil was calculated 1.20 Mg m-3.”
Point 13: Page 3, line 135: Table 1 should be placed in the results and discussion section not in materials and methods. Please correct.
Response 13: The Table 1 is only the "Chemical composition of manure and digestate used in the experiment". This is usefull information for describes the experimental conditions. Therefore, we leave it in the "Material and methods" section. Especially since the other three reviewers did not suggest such changes.
Point 14: Page 4, line 170: This section is very long and difficult to read because of the lack of sub-headings. For example, from lines 171 to 193, the paragraph talks about the comparison of cow liquid manure and digestate, and it has no relevance to the title. This paragraph can independently have separate sub-headings.
Response 14: Yes, you are right, this paragraph can independently have separate sub-headings. However, we have organized the "Results and Discussion" section in a logical sequence: i.e. what significance does digestate have for soil properties (subsection 3.1.), plants – spring wheat (subsection 3.2.) and leaching of nutrients (subsection 3.3.).
Point 15: Page 4. Line 147: Based on this sentence, the authors have determined inorganic nitrogen but reported only ammonium content in table 1. Please correct and clarify this contradiction.
Response 15: It is corrected and in the Table 1. Is provided additional information about nitrate nitrogen
Point 16: Page 4. Line 148: The authors should briefly mention the method used for phosphorus and potassium measurement.
Response 16: It is corrected.
Point 17: Page 4, Line 149: Based on this section, the authors should add the applied method for soil texture measurement earlier in the material and methods section. In addition, the content of soil organic matter should be reported.
Response 17: It is corrected and added the applied method for soil texture measurement
Point 18: Line 14, 317, and 165: The authors have reported the content of total pigments and carotenoids in the aforementioned lines, but no explanation has been provided about the applied methods. The relevant method should be presented in the materials and methods section.
Response 18: Not only chlorophyll (a, b) and carotenoid content were determined using the Holm–Wettstein methodology. Now is corrected and wrotte “Chlorophyll (a, b), carotenoid and total pigments content in leaves were determined at BBCH 25 of spring wheat, using the Holm–Wettstein methodology [26,27].”
Point 19: Line 168: There is a contradiction between line 168 and line 104, Based on line 104, the test was performed in 4 replicates, and authors should correct and clarify this section.
Response 19: Sorry, but between line 168 and line 104 isn't a contradiction, because there are two different things. In the lines 103-104 is written "Treatments were tested in four replicates in a completely randomized block design." and in line 168 is written "The article presents averages of data from two experiments repeated over time." It means, that experiment was repeted two time with all treatment which were tested in four replicates in a completely randomized block design. We have corrected and added information "Treatments were tested in four replicates in a completely randomized block design. Two experiments were conducted and repeated over time in 2019 and 2020." It might be more clearly.
Point 20: Page 4, Line 171-184: This section should be revised and be better organized. The authors should first report the experimental results of this study and improve this section with more involved mechanisms. Then, compare the results of the present research with some similar studies, which is done before
Response 20: This section we have corrected and included new section 3.1. Comparison of the chemical composition of liquid cow manure and digestate
Point 20: Page 5, Line 219: Provide a reference to this sentence, please
Response 20: It is corrected and included these references:
Wang, J.; Zhang, J.; Muller, C.; Cai, Z. Temperature sensitivity of gross N transformation rates in an alpine meadow on the Qinghai-Tibetan Plateau. J. Soils Sediments 2017, 17, 423–431. doi.org/10.1007/s11368-016-1530-2
Zhang, S.; Zheng, Q.; Noll, L.; Hu, Y.; Wanek, W. Environmental effects on soil microbial nitrogen use efficiency are controlled by allocation of organic nitrogen to microbial growth and regulate gross N mineralization. Soil Biol Biochem. 2019, 135, 304–315. doi:10.1016/j.soilbio.2019.05.019
Point 21: Figures 1, 2, 3, and 4: The determinations of these figures have to include the error bars
Response 21: Sorry, but error bars should only be included when there are big differences between variants, but they are not significant. High variance of the data would justify the absence of significant differences. In this case, significant differences was identified and error bars would be excess of information. In the presented figures and tables, all data are evaluated by the analysis of variance method (ANOVA), and significant differences are marked with different letters. We not found information that the data must be evaluated using several statistical methods.
Thank you!

Round 2
Reviewer 1 Report
The authors have address all the comments and now it is able to publish in the journal.
Reviewer 4 Report
I think it is nice MS and lies within the scope of the Journal.